# Subgraph-Aware Training of Language Models for Knowledge Graph Completion Using Structure-Aware Contrastive Learning

## Abstract

Fine-tuning pre-trained language models (PLMs) has recently shown a potential to improve knowledge graph completion (KGC). However, most PLM-based methods focus solely on encoding textual information, neglecting the long-tailed nature of knowledge graphs and their various topological structures, e.g., subgraphs, shortest paths, and degrees. We claim that this is a major obstacle to achieving higher accuracy of PLMs for KGC. To this end, we propose a Subgraph-Aware Training framework for KGC (SATKGC) with two ideas: (i) subgraph-aware mini-batching to encourage hard negative sampling and to mitigate an imbalance in the frequency of entity occurrences during training, and (ii) new contrastive learning to focus more on harder in-batch negative triples and harder positive triples in terms of the structural properties of the knowledge graph. To the best of our knowledge, this is the first study to comprehensively incorporate the structural inductive bias of the knowledge graph into fine-tuning PLMs. Extensive experiments on three KGC benchmarks demonstrate the superiority of SATKGC. Our code is available.[1]

## CCS Concepts

• **Computing methodologies → Knowledge representation and reasoning**.

## Keywords

Knowledge Graph Completion, Contrastive Learning, Hard Negative Sampling

**ACM Reference Format:**

Anonymous Author(s). 2025. Subgraph-Aware Training of Language Models for Knowledge Graph Completion Using Structure-Aware Contrastive Learning. In *Proceedings of the ACM Web Conference (WWW '25)*. ACM, New York, NY, USA, 15 pages. https://doi.org/XXXXXXX.XXXXXXX

## 1 INTRODUCTION

Factual sentences, e.g., Leonardo da Vinci painted Mona Lisa, can be represented as *entities*, and *relations* between the entities. Knowledge graphs treat the entities (e.g., Leonardo da Vinci, Mona Lisa) as nodes, and the relations (e.g., painted) as edges. Each edge and its endpoints are denoted as a triple $(h, r, t)$, where $h, r$, and $t$ are a head entity, a relation, and a tail entity respectively. Since KGs can

[1]https://anonymous.4open.science/r/SATKGC-61B0/README.md

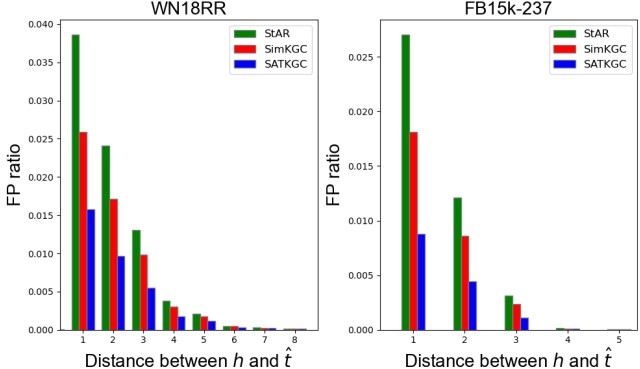

**Figure 1: False positive (FP) ratio against the distance (i.e., length of the shortest path) between head and tail of a FP triple in KG across different text-based methods.**

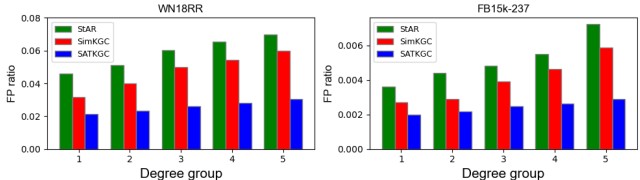

**Figure 2: False positive (FP) ratio against the degree of tail for a FP triple across different text-based methods.**

represent complex relations between entities, they serve as key components for knowledge-intensive applications [12, 16, 20, 43, 49].

Despite their applicability, factual relations may be missing in incomplete real-world KGs, and these relations can be inferred from existing facts in the KGs. Hence, the task of knowledge graph completion (KGC) has become an active research topic [16]. Given an incomplete triple $(h, r, ?)$, this task is to predict the correct tail $t$. A true triple $(h, r, t)$ in a KG and a false triple $(h, r, \hat{t})$ which does not exist in the KG are called positive and negative, respectively. A negative triple difficult for a KGC method to distinguish from its corresponding positive triple is regarded as a hard negative triple.

Existing KGC methods are categorized into two approaches. An embedding-based approach learns embeddings of entities in continuous vector spaces, but ignores contextualized text information in KGs, thus being inapplicable to entities and relations unseen in training [2, 3]. A text-based approach, based on pretrained language models (PLMs), learns textual representations of KGs, but suffers from a lack of structural knowledge inherent in KGs [39]. Moreover, most methods in this approach fail to account for the long-tailed distribution of entities in KGs, though they can perform KGC even in the *inductive setting*[2].

[2]In the inductive setting for KGC, entities in the test set never appear in the training set.

Meanwhile, contrastive learning has become a key component of representation learning [9, 17, 29, 39], but an important aspect of contrastive learning, i.e., the effect of hard negatives, has so far been underexplored in KGC.

In this paper, we empirically validate significant relationships between the structural inductive bias of KGs and the performance of the PLM-based KGC methods. To demonstrate the limitations of language models exhibiting competitive performance such as SimKGC [39] and StAR [38], we investigate the characteristics of false positive (FP) triples, i.e., false triples ranked higher than a true triple by the models, on two well-known datasets WN18RR and FB15k-237. Our analysis draws two observations.

First, the closer the tail and head of a false triple are to each other in the KG, the more likely the false triple is to be ranked higher than the corresponding true triple. Figure 1 illustrates the distribution of distance, i.e., the length of the shortest path, between the head and tail of a FP triple, where $y$-axis represents the FP ratio[3] for each distance. For StAR and SimKGC, the FP ratio dramatically grows as the distance decreases (see green and red bars in Figure 1). These findings highlight the importance of considering the proximity between two entities in a triple to distinguish between true and false triples.

Second, the higher the degree of the tail in a false triple is, the more likely that triple is to be ranked higher than the corresponding true triple. Figure 2 illustrates the distribution for the degree of tails of FP triples. They are sorted in ascending order of their degrees, and then are divided into five groups such that each group contains an equal number of distinct degrees. The $y$-axis represents the FP ratio[4] in each degree group. The FP ratio for StAR and SimKGC increases as the degree of the tail grows (see green and red bars in Figure 2). This indicates that the existing text-based methods have difficulty in predicting correct tails for missing triples with the high-degree tails. Hence, the degree can be taken into account to enhance the performance of language models.

In this paper, we tackle the above two phenomena[5], thereby significantly reducing the FPs compared to the existing methods (see blue bars in Figures 1 and 2). For this, we hypothesize that incorporating the structural inductive bias of KGs into hard negative sampling and fine-tuning PLMs leads to a major breakthrough in learning comprehensive representations of KGs.

To this end, we propose a **subgraph-aware PLM training framework for KGC (SATKGC)** with three novel techniques: (i) we sample subgraphs of the KG, and treat triples of each subgraph as a mini-batch to encourage hard negative sampling and to alleviate the negative effects of long-tailed distribution of the entity frequencies during training; (ii) we fine-tune PLMs via a novel contrastive learning method that focuses more on harder negative triples, i.e., negative triples whose heads and tails are close to each other in the KG, induced by topological bias in the KG; and (iii) we propose balanced mini-batch loss that mitigates the gap between the long-tailed distribution of the training set and a nearly uniform distribution of each mini-batch. To sum up, we make three contributions.

- We provide key insights that the topological structure of KGs is closely related to the performance of PLM-based KGC methods.
- We propose a new training strategy for PLMs, which not only effectively samples hard negatives from a subgraph but also visits all entities in the KG nearly equally in training, as opposed to the existing PLM-based KGC methods. Based on the structural properties of KGs, our contrastive learning enables PLMs to pay attention to difficult negative triples in KGs, and our mini-batch training eliminates the discrepancy between the distributions of a training set and a mini-batch.
- We conduct extensive experiments on three KGC benchmarks to demonstrate the superiority of SATKGC over existing KGC methods.

## 2 RELATED WORK

An **embedding-based approach** maps complex and structured knowledge into low-dimensional spaces. This approach computes the plausibility of a triple using translational scoring functions on the embeddings of the triple's head, relation, and tail [3, 10, 32, 53], e.g., $h + r \approx t$, or semantic matching functions which match latent semantics of entities and relations [2, 23, 36, 44]. KG-Mixup [30] proposes to create synthetic triples for a triple with a low-degree tail. UniGE [21] utilizes both hierarchical and non-hierarchical structures in KGs. The embedding-based approach exploits the spatial relations of the embeddings, but cannot make use of texts in KGs, i.e., the source of semantic relations.

In contrast, a **text-based approach** learns contextualized representations of the textual contents of entities and relations by leveraging PLMs [4, 7, 18, 39, 40, 42, 47]. Recently, a sequence-to-sequence model for KGC [48] highlights the growing trend of leveraging natural language generation models. With the rise of large language models (LLMs), their application in KGs has significantly increased [52]. Despite these advancements, PLMs often lack awareness of the structural inductive bias inherent in KGs.

A few attempts have been made to utilize the above two approaches at once. StAR [38] proposes an ensemble model incorporating an output of a Transformer encoder [37] with a triple score produced by RotatE [32]. CSProm-KG [5] trains KG embeddings through the soft prompt for a PLM. Nevertheless, the integration of structural and textual information in a KG in training has not yet been fully realized.

**Contrastive learning**, shown to be effective in various fields [8, 9, 11, 41, 51], has recently emerged as a promising approach in the context of KGC [27, 39, 46]. KRACL [34] introduces contrastive learning into an embedding-based method, particularly through the use of a graph neural network (GNN), while HaSa [50] applies contrastive learning to a PLM. HaSa aims to sample hard negatives which are unlikely to be false negatives by selecting tails of a negative triple based on the number of 1- and 2-hop neighbors of its head entity, i.e., the greater the number, the lower the probability that the tail is included in the false negative. In contrast, our contrastive learning method estimates the difficulty of negative triples by utilizing various length of the shortest path between their

---

[3](the number of FPs with a specific head-to-tail distance) / (the number of pairs of entities in the KG with the same distance)

[4](the average number of FPs whose tail's degree falls into each group) / (the number of entities in the KG whose degree falls into each group)

[5]The trends in Figures 1 and 2 are also confirmed on Wikidata5M.

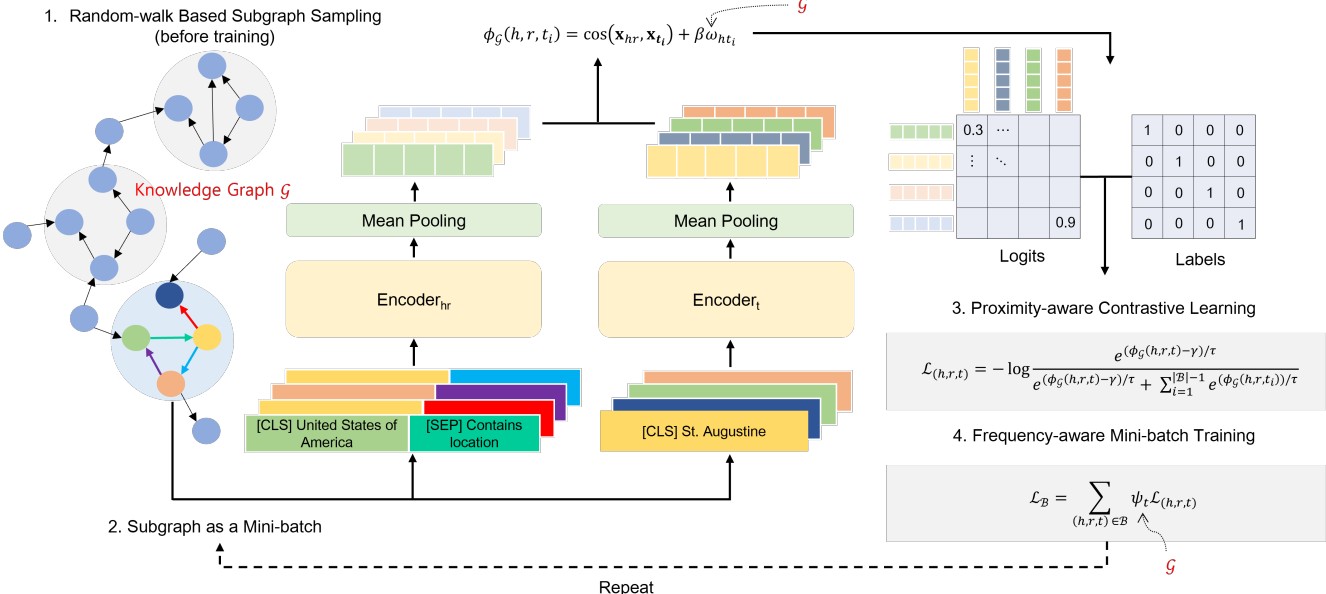

**Figure 3: Overview of the proposed training framework, which consists of: (i) Random-walk Based Subgraph Sampling (before training); (ii) Subgraph as a Mini-batch; (iii) Proximity-aware Contrastive Learning; (iv) Frequency-aware Mini-batch Training.**

head and tail, while simultaneously mitigating the adverse effects caused by the long-tailed distribution in KGs. The long-tailed distribution problem, which reflects class imbalance in data, has been extensively studied in other domains, particularly computer vision, where numerous efforts have been made to address this challenge [15, 19, 28, 33, 54].

Random walk with restart (RWR) [25] and its extension, biased random walk with restart (BRWR), have been employed in various domains such as node representation learning [26] and graph traversals [1]. In BRWR, a random walker performs random walks in a graph from the source node. For each iteration, the walker moves from the current node $u$ to either (i) source with a probability of $p_r$, or (ii) one of the neighbors of $u$ with a probability of $1 - p_r$, where $p_r$ is a hyperparameter. In case (ii), the probability of selecting one of the neighbors is decided by a domain-specific probability distribution, whereas one is selected uniformly at random in RWR. To our knowledge, we are the first to extract a subgraph of KG via BRWR to utilize the subgraph as a mini-batch during training.

## 3 PRELIMINARY

Let $\mathcal{G} = (\mathcal{E}, \mathcal{R}, \mathcal{T})$ denote a KG in which $\mathcal{E}$ and $\mathcal{R}$ represent a set of entities and relations, respectively, and $\mathcal{T} = \{(h, r, t)|h, t \in \mathcal{E}, r \in \mathcal{R}\}$ is a set of triples where $h$, $r$, and $t$ are a head entity, a relation, and a tail entity, respectively.

**Problem Definition.** Given a head $h$ and a relation $r$, the task of knowledge graph completion (KGC) aims to find the most accurate tail $t$ such that a new triple $(h, r, t)$ is plausible in $\mathcal{G}$.

**InfoNCE Loss.** InfoNCE loss [24] has been widely employed to learn the representations for audio, images, natural language, and even for KGs [39, 50]. Given a set $X = \{t_1, t_2, ..., t_n\}$ of $n$ tails containing one positive sample from $p(t_j|h, r)$ and $n - 1$ negative samples

from the proposal distribution $p(t_j)$, InfoNCE loss [39] for KGC is defined as:

$$\mathcal{L}_X = \mathbb{E}_X \left[ -\log \frac{\exp(\phi(h, r, t))}{\sum_{t_j \in X} \exp(\phi(h, r, t_j))} \right] \quad (1)$$

$$\phi(h, r, t) = \cos(\mathbf{x}_{hr}, \mathbf{x}_t) \quad (2)$$

where a scoring function $\phi(h, r, t)$ is defined as the cosine similarity between two representations $\mathbf{x}_{hr}$ for $h$ and $t$, and $\mathbf{x}_t$ for $t$.

## 4 METHOD

In this section, we describe a novel PLM fine-tuning framework for KGC. Figure 3 illustrates the overview of our framework for a given KG. First, for every triple, a subgraph is extracted around that triple from the KG before training (Section 4.1). During training, we keep track of the number of visits for every triple. For each iteration, a subgraph is selected based on that number, and then all forward and inverse triples in the subgraph are fetched as a mini-batch $\mathcal{B}$ to the language model (Section 4.2). We adopt the bi-encoder architecture [39] with two pre-trained MPNets [31] as a backbone. Specifically, $\text{Encoder}_{hr}$ and $\text{Encoder}_t$ take the text, i.e., name and description, of $(h, r)$ and $t$ as input, and produce their embeddings $\mathbf{x}_{hr}$ and $\mathbf{x}_t$ respectively. We then calculate the cosine similarity between $\mathbf{x}_{hr}$ and $\mathbf{x}_t$ for every $(h, r)$ and $t$ in the mini-batch, and perform new contrastive learning equipped with two topology-aware factors (Section 4.3 and 4.4). The details of the input format are provided in Appendix A, and the model inference is described in Appendix B.

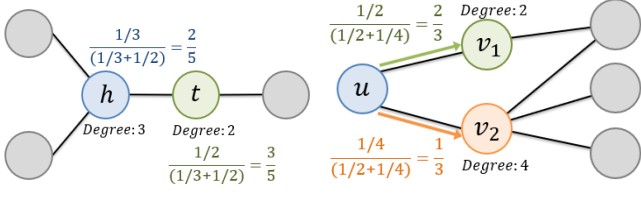

**(a)**                                          **(b)**

**Figure 4: Example of BRWR-based subgraph sampling; (a) probability of selecting start entity $s$ between $h$ and $t$ of a center triple, where $t$ with a lower degree is more likely to be $s$ than $h$; (b) probability of selecting a neighbor of current entity $u$. A random walker is more likely to move to $v_1$ than to $v_2$ with its degree larger than $v_1$.**

## 4.1 Random-walk Based Subgraph Sampling

We aim to extract subgraphs from the KG to treat all the triples in the extracted subgraph as a mini-batch for training. For each triple in the KG, therefore, we perform BRWR starting from that triple called a center triple, and the triples visited by BRWR compose the subgraph before training as follows: (i) either head $h$ or tail $t$ of the center triple is selected as the start entity $s$ based on an inverse degree distribution of $h$ and $t$, i.e., $\frac{|N(v)|^{-1}}{|N(h)|^{-1}+|N(t)|^{-1}}$, where $v \in \{h, t\}$ and $N(v)$ denotes a set of $v$'s neighbors; (ii) next, we perform BRWR from $s$ until we sample $M$ triples where $M$ is a predefined maximum number (e.g., 10,000). For each iteration in BRWR, a random walker moves from the current node to either $s$ with a probability of $p_r$ or one of the neighbors of the current node with a probability of $1 - p_r$. We define the probability of selecting one of $u$'s neighbors $v \in N(u)$ as $p_v = \frac{|N(v)|^{-1}}{\sum_{v_j \in N(u)} |N(v_j)|^{-1}}$, which is a normalized inverse degree distribution of the neighbors. Figures 4a and 4b show the running example of step (i) and an iteration of step (ii).

In this manner, giving more chances for an entity with a lower degree to engage in training alleviates the skewed frequency of entity occurrences throughout training, which in turn enhances the KGC performance. This claim will be validated in Section 5.5.

## 4.2 Subgraph as a Mini-batch

We now present a way to select one of the sampled subgraphs, and utilize that subgraph as a mini-batch, dubbed **S**ubgraph **a**s a **M**ini-batch (**SaaM**). For every iteration, we select the subgraph whose center triple has been the least frequently visited, to prioritize unpopular and peripheral triples. For this, we count the number of visits for all triples in the training set $\mathcal{T}$ throughout the training process. The rationale behind this selection will be elaborated in Section 5.5. Next, we randomly select $|\mathcal{B}|/2$ triples from the subgraph, and feed to the bi-encoders a mini-batch $\mathcal{B}$ of the selected triples $(h, r, t)$ and their inverse triples $(t, r^{-1}, h)$. For every positive triple $(h, r, t) \in \mathcal{B}$, we obtain negative triples $(h, r, \hat{t})$ with $t$ replaced by $|\mathcal{B}| - 1$ tails $\hat{t}$ of the other triples in $\mathcal{B}$. As per our observation in Figure 1, these negative triples are likely to be hard negatives, which will facilitate contrastive learning. As a result, we end up with iterating the above process, i.e., selecting the subgraph

and feeding the triples in that subgraph, $|\mathcal{T}|/|\mathcal{B}|$ times for each epoch.

## 4.3 Proximity-aware Contrastive Learning

Most PLMs for KGC overlook capturing the proximity between two entities in a negative triple in the KG, though they capture semantic relations within the text of triples, as described in the first observation of Section 1. For effective contrastive learning for these methods, we incorporate a topological characteristic, i.e., the proximity between two entities, of the KG into InfoNCE loss with additive margin [6, 45] by measuring how hard a negative triple is in the KG. For each positive triple $(h, r, t)$ in a mini-batch $\mathcal{B}$, we propose loss $\mathcal{L}_{(h,r,t)}$ below based on our aforementioned observation, i.e., entities close to each other are more likely to be related than entities far away from each other:

$$\mathcal{L}_{(h,r,t)} = -\log \frac{\exp(\frac{\phi_{\mathcal{G}}(h,r,t)-\gamma}{\tau})}{\exp(\frac{\phi_{\mathcal{G}}(h,r,t)-\gamma}{\tau}) + \sum_{i=1}^{|B|-1} \exp(\frac{\phi_{\mathcal{G}}(h,r,t_i)}{\tau})} \quad (3)$$

$$\phi_{\mathcal{G}}(h, r, t_i) = \cos(\mathbf{x}_{hr}, \mathbf{x}_{t_i}) + \beta\omega_{ht_i} \quad (4)$$

where $\gamma$ is an additive margin, a temperature parameter $\tau$ adjusts the importance of negatives, a structural hardness factor $\omega_{ht_i}$ stands for how hard a negative triple $(h, r, t_i)$ is in terms of the structural relation between $h$ and $t_i$ in $\mathcal{G}$, and $\beta$ is a trainable parameter that adjusts the relative importance of $\omega_{ht_i}$. We define $\omega_{ht_i}$ as the reciprocal of the distance (i.e., length of the shortest path) between $h$ and $t_i$ to impose a larger $\omega_{ht_i}$ to the negative triple with a shorter distance between $h$ and $t_i$ in $\mathcal{G}$, serving as a hard negative triple.

Since computing the exact distance between every head and every tail in $\mathcal{B}$ may spend considerable time, we calculate the approximate distance between $h$ and $t_i$, i.e., the multiplication of two distances: (d1) the distance between $h$ and head $h_c$ of the center triple of $\mathcal{B}$, and (d2) the distance between $t$ and $h_c$. Thus, the distance from $h_c$ to every entity in $\mathcal{B}$ is pre-computed before training, the multiplication between the two distances is performed in parallel during training, requiring a minimal computational overhead.[6]

## 4.4 Frequency-aware Mini-batch Training

For many triples with the same head in the KG, varying only relation $r$ in $(h, r, ?)$ may lead to different correct tails with various semantic contexts. The text-based KGC methods may find it difficult to predict the correct tail for these triples, as their head-relation encoders may generate less diverse embeddings for $(h, r)$ due to much shorter text of relation $r$ than entity $h$.

Furthermore, the frequency of every entity that occurs in a set $\mathcal{T}$ of triples varies; this frequency distribution follows the power law. However, the text-based methods have shown difficulties in addressing the discrepancy between this long-tailed distribution of the KG and the distribution of a mini-batch. Figure 5 shows the frequency distributions of the original KG, 100 randomly-sampled mini-batches from $\mathcal{T}$, and those from SaaM, where $y$-axis denotes the frequency ratio of entity occurrences, and entities in $x$-axis are

---

[6]We conducted experiments using the exact distance and different approximate distances, e.g., the sum of (d1) and (d2), but the performance gap between all the methods is very marginal.

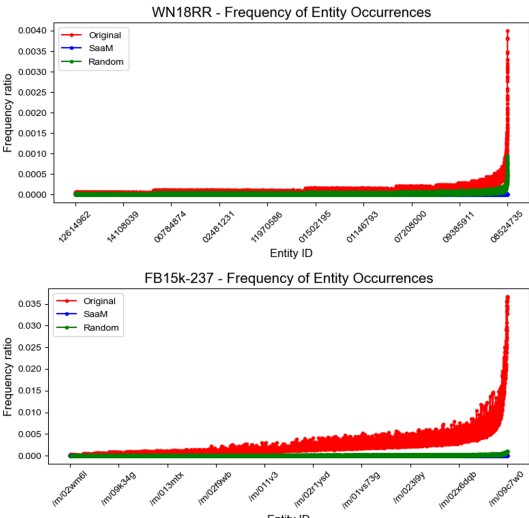

**Figure 5: Frequency distributions of entities for original KG, 100 mini-batches randomly sampled from $\mathcal{T}$, and those randomly sampled by SaaM on WN18RR and FB15k-237. The entities are sorted in the ascending order of their degrees.**

sorted in the ascending order of their degrees in KG. Given $h$ and $t$, in fact, KGC can be regarded as the multi-class classification task to predict a correct tail $t$. From this perspective, severe class imbalance originating from the long-tail distribution (red lines) in the KG may result in the less-skewed distribution (green lines) for 100 randomly-sampled mini-batches from $\mathcal{T}$, due to a limited number of entities in the mini-batches. In addition, we observe the nearly uniform distribution for 100 mini-batches selected by SaaM (blue lines), since SaaM is likely to give preference to low-degree entities. Therefore, we apply the importance sampling scheme to associate the expected error of $\mathcal{L}_{(h,r,t)}$ with the long-tailed original domain of the KG:

$$\mathbb{E}_{p_o(h,r,t)} \left[ \mathcal{L}(h,r,t) \right] = \mathbb{E}_{p_s(h,r,t)} \left[ \frac{p_o(h,r,t)}{p_s(h,r,t)} \mathcal{L}_{(h,r,t)} \right]$$
$$= \mathbb{E}_{p_s(h,r,t)} \left[ \frac{p_o(t)p(h,r|t)}{p_s(t)p(h,r|t)} \mathcal{L}_{(h,r,t)} \right]$$
$$:= \mathbb{E}_{p_s(h,r,t)} \left[ \psi_{\mathcal{G}}(t) \mathcal{L}_{(h,r,t)} \right]$$

where $p_s(h,r,t)$ and $p_o(h,r,t)$ are probabilities of sampling $(h,r,t)$ by SaaM and randomly in $\mathcal{T}$ respectively. A weighting factor $\psi_{\mathcal{G}}(t)$ of tail $t$ denotes $p_o(t)/p_s(t)$. Assume the degree-proportional $p_o(t) = |N_t|/2|\mathcal{T}|$ and the nearly-uniform distribution $p_s(t) = 1/|\mathcal{E}|$ where $|\mathcal{T}| = d_{avg}|\mathcal{E}|/2$ with $d_{avg}$ being the average number of neighbors for every entity. Consequently, $p_o(t)/p_s(t) = |N_t|/d_{avg} \propto |N_t|$.

To encourage PLMs to more sensitively adapt to varying relations for many triples with the identical head and to eliminate the discrepancy of the two distributions, we propose to reweight the importance of each sample loss via $\psi_{\mathcal{G}}(t)$ above, so the frequency distribution of $\mathcal{B}$ becomes close to that of $\mathcal{T}$. For each triple in a mini-batch $\mathcal{B}$ in SaaM, mini-batch loss $\mathcal{L}_{\mathcal{B}}$ is defined as:

$$\mathcal{L}_{\mathcal{B}} = \sum_{(h,r,t)\in\mathcal{B}} \psi_{\mathcal{G}}(t) \mathcal{L}_{(h,r,t)} \tag{5}$$

| dataset | #entity | #relation | #train | #valid | #test |
|---|---|---|---|---|---|
| WN18RR | 40,943 | 11 | 86,835 | 3,034 | 3,134 |
| FB15k-237 | 14,541 | 237 | 272,115 | 17,535 | 20,466 |
| Wikidata5M-Trans | 4,594,485 | 822 | 20,614,279 | 5,163 | 5,163 |
| Wikidata5M-Ind | 4,579,609 | 822 | 20,496,514 | 6,699 | 6,894 |

**Table 1: Statistics of datasets.**

where $\psi_{\mathcal{G}}(t)$ is defined as $log(|N_t| + 1)$ where $N_t$ is a set of $t$'s neighbors in the KG[7]. To sum up, $\psi_{\mathcal{G}}(t)$ ensures that the triples with larger-degree tails contribute more significantly to $\mathcal{L}_{\mathcal{B}}$.

## 5 EXPERIMENTS

### 5.1 Experimental Setup

For evaluation we adopt widely-used KG datasets WN18RR, FB15k-237 and Wikidata5M. Table 1 shows their statistics, and more details are provided in Appendix D.

For every incomplete triple in the test set, we compute mean reciprocal rank (MRR) and Hits@$k$ where $k \in \{1, 3, 10\}$ as evaluation metrics based on the rank of the correct entity among all the entities in the KG. We use the mean of the forward and backward prediction results as the final performance measure. The hyperparameters of SATKGC are set based on the experimental results in Appendix F. Further implementation details are described in Appendix E.

### 5.2 Main Results

We compare SATKGC with existing embedding-based and text-based approaches. Table 2 shows the results on WN18RR, and FB15k-237, and Table 3 shows the results on Wikidata. Due to the page limit, we present comparison with recent models in these two tables. Additional comparison results can be found in Appendices K and L.

SATKGC denotes the bi-encoder architectures trained by our learning framework in Figure 3. SATKGC consistently outperforms all the existing methods on all the datasets. SATKGC demonstrates significantly higher MRR and Hits@1 than other baselines, with Hits@1 improving by 5.03% on WN18RR and 5.28% on FB15k-237 compared to the existing state-of-the-art models.

As shown in Table 3, SATKGC demonstrates its applicability to large-scale KGs, and achieves strong performance in both inductive and transductive settings.[8] SATKGC shows an improvement of 7.42% in MRR and 10.84% in Hits@1 compared to the existing state-of-the-art model on Wikidata5M-Ind. On Wikidata5M-Trans, SATKGC achieves an improvement of 13.97% in MRR and 16.93% in Hits@1 over the previous best-performing model.[9] In the transductive setting, performance degrades in the order of WN18RR, Wikidata5M-Trans, and FB15k-237, showing that a higher average degree of entities tends to negatively affect performance. A more detailed analysis of performance differences across the dataset is described in Appendix J.

---

[7]Applying logarithm can prevent $\psi_{\mathcal{G}}(t)$ from growing extremely large for a few entities with a huge $|N_t|$, and adding one can prevent $\psi_{\mathcal{G}}(t)$ from becoming zero when $|N_t| = 1$.

[8]Baselines listed in Table 2 but not in Table 3 could not be evaluated on Wikidata5M due to out-of-memory for StAR and CSProm-KG, or unreasonably large training time, i.e., more than 100 hours expected, for the remaining baselines.

[9]Wikidata5M-Ind shows better performance than Wikidata5M-Trans, because a model ranks 7,475 entities in the test set for Wikidata5M-Ind while ranking about 4.6 million entities for Wikidata5M-Trans.

| Approach | WN18RR | | | | FB15k-237 | | | |
|---|---|---|---|---|---|---|---|---|
| | MRR | Hits@1 | Hits@3 | Hits@10 | MRR | Hits@1 | Hits@3 | Hits@10 |
| *Embedding-based approach* | | | | | | | | |
| TuckER [2] | 0.466 | 0.432 | 0.478 | 0.518 | 0.361 | 0.265 | 0.391 | 0.538 |
| RotatE [32] | 0.471 | 0.421 | 0.490 | 0.568 | 0.335 | 0.243 | 0.374 | 0.529 |
| KGTuner [53] | 0.481 | 0.438 | 0.499 | 0.556 | 0.345 | 0.252 | 0.381 | 0.534 |
| KG-Mixup [30] | 0.488 | 0.443 | 0.505 | 0.541 | 0.359 | 0.265 | 0.395 | 0.547 |
| UniGE[21]† | 0.491 | 0.447 | 0.512 | 0.563 | 0.343 | 0.257 | 0.375 | 0.523 |
| CompoundE [10] | 0.492 | 0.452 | 0.510 | 0.570 | 0.350 | 0.262 | 0.390 | 0.547 |
| KRACL [34] | 0.529 | 0.480 | 0.539 | 0.621 | 0.360 | 0.261 | 0.393 | 0.548 |
| CSProm-KG [5] | 0.569 | 0.520 | 0.590 | 0.675 | 0.355 | 0.261 | 0.389 | 0.531 |
| *Text-based approach* | | | | | | | | |
| StAR [38] | 0.398 | 0.238 | 0.487 | 0.698 | 0.288 | 0.195 | 0.313 | 0.480 |
| HaSa [50] | 0.535 | 0.446 | 0.587 | 0.711 | 0.301 | 0.218 | 0.325 | 0.482 |
| KG-S2S [4] | 0.572 | 0.529 | 0.595 | 0.663 | 0.337 | 0.255 | 0.374 | 0.496 |
| SimKGC [39] | 0.671 | 0.580 | 0.729 | 0.811 | 0.340 | 0.252 | 0.365 | 0.515 |
| GHN [27]‡ | 0.678 | 0.596 | 0.719 | 0.821 | 0.339 | 0.251 | 0.364 | 0.518 |
| SATKGC (*w/o* SaaM) | 0.673 | 0.595 | 0.728 | 0.813 | 0.349 | 0.256 | 0.367 | 0.520 |
| SATKGC (*w/o* PCL, FMT) | 0.676 | 0.608 | 0.722 | 0.820 | 0.355 | 0.261 | 0.386 | 0.537 |
| SATKGC (*w/o* FMT) | 0.680 | 0.611 | 0.729 | 0.823 | 0.351 | 0.265 | 0.389 | 0.541 |
| SATKGC (*w/o* PCL) | 0.686 | 0.623 | 0.740 | 0.827 | 0.366 | 0.272 | 0.396 | 0.545 |
| SATKGC | **0.694** | **0.626** | **0.743** | **0.833** | **0.370** | **0.279** | **0.403** | **0.550** |

**Table 2: KGC results for the WN18RR, and FB15k-237 datasets. "PCL" and "FMT" refer to Proximity-aware Contrastive Learning and Frequency-aware Mini-batch training respectively. The best and second-best performances are denoted in bold and underlined respectively. †: numbers are from Liu et al. [21] ‡: numbers are from Qiao et al. [27].**

| Approach | Wikidata5M-Trans | | | | Wikidata5M-Ind | | | |
|---|---|---|---|---|---|---|---|---|
| | MRR | Hits@1 | Hits@3 | Hits@10 | MRR | Hits@1 | Hits@3 | Hits@10 |
| *Embedding-based approach* | | | | | | | | |
| TransE [3] | 0.253 | 0.170 | 0.311 | 0.392 | - | - | - | - |
| TuckER [2] | 0.285 | 0.241 | 0.314 | 0.377 | - | - | - | - |
| RotatE [32] | 0.290 | 0.234 | 0.322 | 0.390 | - | - | - | - |
| KGTuner [53] | 0.305 | 0.243 | 0.331 | 0.397 | - | - | - | - |
| *Text-based approach* | | | | | | | | |
| DKRL [42] | 0.160 | 0.120 | 0.181 | 0.229 | 0.231 | 0.059 | 0.320 | 0.546 |
| KEPLER [40] | 0.212 | 0.175 | 0.221 | 0.276 | 0.403 | 0.225 | 0.517 | 0.725 |
| BLP-ComplEx [7] | - | - | - | - | 0.491 | 0.261 | 0.670 | 0.881 |
| BLP-SimplE [7] | - | - | - | - | 0.490 | 0.283 | 0.641 | 0.868 |
| SimKGC [39] | 0.358 | 0.313 | 0.376 | 0.441 | 0.714 | 0.609 | 0.785 | 0.917 |
| SATKGC | **0.408** | **0.366** | **0.425** | **0.479** | **0.767** | **0.675** | **0.815** | **0.931** |

**Table 3: KGC results for Wikidata5M-Trans (transductive setting) and Wikidata5M-Ind (inductive setting). The results for embedding-based approach on Wikidata5M-ind are missing as they cannot be used in the inductive setting. Additionally, BLP-ComplEx [7] and BLP-SimplE [7] results on Wikidata5M-Trans are missing because they are inherently targeted for inductive KGC.**

| | FB15K-237N | | | |
|---|---|---|---|---|
| Models | MRR | Hits@1 | Hits@3 | Hits@10 |
| KoPA [52] | 0.483 | 0.344 | 0.559 | 0.721 |
| SATKGC | **0.569** | **0.450** | **0.637** | **0.792** |

**Table 4: KGC results on FB15K-237N. A correct tail is ranked among 1,000 randomly-selected entities due to the long inference time of KoPA.**

To compare our framework employing MPNets with a LLM-based model, we evaluate the KGC performance of KoPA [52], which

adopts Alpaca [35] fine-tuned with LoRA [14] as its backbone, on the FB15-237N dataset [52].[10] Since KoPA cannot perform inference for all queries within a reasonable time, i.e., 111 hours expected, we rank the correct tail among 1,000 randomly selected entities for both KoPA and SATKGC. As shown in Table 4, SATKGC outperforms KoPA on all metrics, demonstrating that LLMs do not necessarily produce superior results on KGC.

---

[10]We adopt FB15k-237N in Table 4 to compare our model with KoPA, because the official implementation of KoPA provides the pretrained model parameters for not FB15k-237 but FB15K-237N.

| WN18RR | | | | | |
|---|---|---|---|---|---|
| Encoders | MRR | Hits@1 | Hits@3 | Hits@10 | Parameters |
| MPNet | 0.693 | 0.626 | 0.747 | 0.833 | 220M |
| BERT-base | 0.689 | 0.621 | 0.731 | 0.820 | 220M |
| DeBERTa-base | 0.689 | 0.631 | 0.736 | 0.832 | 280M |
| BERT-large | 0.685 | 0.619 | 0.723 | 0.817 | 680M |
| DeBERTa-large | **0.706** | **0.638** | **0.759** | **0.854** | 800M |
| FB15k-237 | | | | | |
| Encoders | MRR | Hits@1 | Hits@3 | Hits@10 | Parameters |
| MPNet | **0.370** | **0.279** | **0.403** | **0.550** | 220M |
| BERT-base | 0.366 | 0.273 | 0.400 | 0.546 | 220M |
| DeBERTa-base | 0.359 | 0.274 | 0.399 | 0.545 | 280M |
| BERT-large | 0.339 | 0.268 | 0.378 | 0.532 | 680M |
| DeBERTa-large | 0.345 | 0.272 | 0.389 | 0.540 | 800M |

**Table 5: Performance comparison for different encoders.**

## 5.3 Ablation Study

To demonstrate the contribution of each component of our method, we compare the results across five different settings, including SATKGC, as shown in Table 2. In SATKGC (*w/o* SaaM), instead of using SaaM, a mini-batch of triples is randomly selected without replacement, while both Proximity-aware Contrastive Learning and Frequency-aware Mini-batch training are applied. SATKGC (*w/o* PCL, FMT) refers to applying only SaaM and using the original InfoNCE loss. SATKGC (*w/o* FMT) applies both SaaM and Proximity-aware Contrastive Learning, and SATKGC (*w/o* PCL) applies both SaaM and Frequency-aware Mini-batch Training. The results show that SATKGC (*w/o* SaaM) performs worse than SATKGC, indicating that substituting SaaM with random sampling significantly hurts the performance. SATKGC (*w/o* PCL, FMT) already achieves higher Hits@1 than other baselines on WN18RR, highlighting that SaaM alone leads to performance improvement[11]. Between SATKGC (*w/o* PCL) and SATKGC (*w/o* FMT), SATKGC (*w/o* PCL) achieves higher performance, indicating that Frequency-aware Mini-batch Training contributes more than Proximity-aware Contrastive Learning.

## 5.4 Performance Across Encoders

To investigate the impact of the encoder architecture and the number of model parameters, we conduct experiments replacing MP-Net in SATKGC with BERT-base, BERT-large, DeBERTa-base, and DeBERTa-large [13]. Table 5 presents the results. SATKGC is highly compatible with different encoders, showing the competitive performance. DeBERTa-large fine-tuned by SATKGC achieves the best performance on WN18RR. In addition, an increase in the number of model parameters may not necessarily result in enhanced performance on KGC, e.g., BERT-large on WN18RR, and BERT-large and DeBERTa-large on FB15k-237 underperform the smaller encoders.

## 5.5 Comparing Subgraph Sampling Methods

We investigate how model performance varies depending on the probability distribution $p_v$ used for neighbor selection in Section 4.1.[12] We compare the performance of SATKGC using $p_v$ in subgraph sampling with two variants, one with $p_v$ replaced by the uniform distribution (dubbed RWR) and the other with $p_v$ replaced by the degree proportional distribution (dubbed BRWR_P). Table

---

[11]Performance improvements are also observed when we apply SaaM to embedding-based RotatE and text-based StAR, as described in Appendix G.
[12]Recall that in our BRWR algorithm, a random walker selects one of the neighbors $v$ of a current node based on the inverse degree distribution $p_v$.

| WN18RR | | | | |
|---|---|---|---|---|
| Methods | MRR | Hits@1 | Hits@3 | Hits@10 |
| RWR | 0.690 | 0.622 | 0.730 | 0.825 |
| BRWR | **0.694** | **0.626** | **0.743** | **0.833** |
| BRWR_P | 0.676 | 0.615 | 0.727 | 0.823 |
| MCMC | 0.687 | 0.609 | 0.736 | 0.825 |
| FB15k-237 | | | | |
| Methods | MRR | Hits@1 | Hits@3 | Hits@10 |
| RWR | 0.358 | 0.269 | 0.385 | 0.534 |
| BRWR | **0.370** | **0.279** | **0.403** | **0.550** |
| BRWR_P | 0.351 | 0.262 | 0.383 | 0.530 |
| MCMC | 0.332 | 0.245 | 0.362 | 0.507 |

**Table 6: Experiments comparing subgraph sampling methods on WN18RR and FB15k-237.**

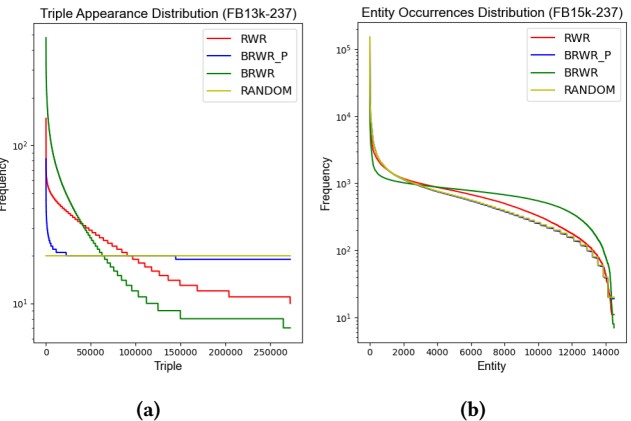

(a)          (b)

**Figure 6: (a) Number of occurrences of triples; (b) number of occurrences of entities. Both are counted throughout the entire training process for RANDOM and the SaaM variants, i.e., RWR, BRWR_P, and BRWR.**

6 shows the results. The three methods mostly outperform existing KGC methods in Hits@1, with BRWR performing best and BRWR_P performing worst. We also employ a Markov chain Monte Carlo (MCMC) based subgraph sampling method [46], referred to as MCMC (see details of MCMC in Appendix C). Note that in Table 2, MCMC outperforms other methods for Hits@1 on WN18RR.

To verify why variations in $p_v$ lead these differences, we conduct further analysis. Figures 6a and 6b show the number of visits of every triple and every entity, respectively, in the training process for RANDOM and the SaaM variants, i.e., RWR, BRWR_P, and BRWR. RANDOM, adopted in all the baselines denotes selecting a mini-batch of triples at random without replacement. Triples and entities are sorted by descending frequency. Figure 6a demonstrates that RANDOM exhibits a uniform distribution, while RWR, BRWR_P, and BRWR display varying degrees of skewness, with BRWR being the most skewed and BRWR_P the least. Figure 6b illustrates that BRWR is the least skewed whereas RANDOM shows the most skewed distribution. BRWR_P, which employs the degree-proportional distribution, extracts many duplicate entities in the subgraphs, leading to the most skewed distribution among the SaaM variants. As a result, a larger skewness in the distribution of number

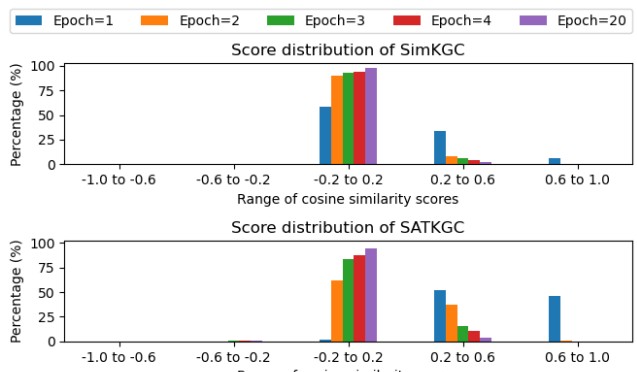

**Figure 7: Percentage of in-batch negative triples on FB15k-237 according to the range of cosine similarity scores predicted by SimKGC and SATKGC for different epochs.**

| Preprocessing Time | | | | |
|---|---|---|---|---|
| Algorithm | WN18RR | FB15k-237 | Wiki5M-Trans | Wiki5M-Ind |
| BRWR | 8m | 12m | 12h | 12h |
| Shortest Path + Degree | 7m | 10m | 7h | 7h |
| Training Time Per Epoch | | | | |
| Model | WN18RR | FB15k-237 | Wiki5M-Trans | Wiki5M-Ind |
| SATKGC | 4m | 6m | 10h | 10h |
| SimKGC [39] | 3m | 5m | 9h | 9h |
| StAR [38] | 1h | 1h 30m | - | - |

**Table 7: The elapsed time required for sampling and the time per epoch during training.**

of visits for triples in turn leads to more equally visiting entities, thus improving the performance.[13]

Further analysis reinforces this finding. In FB15k-237, the average degree of FP triples' tails is 75 for SATKGC and 63 for SimKGC. A smaller portion of low-degree tails for SATKGC than for SimKGC indicates that exposure to more low-degree entities $\hat{t}$ in training helps the model position their embeddings farther from the $(h, r)$ embeddings for negative triples $(h, r, \hat{t})$, as SATKGC visits low-degree entities more often during training than RANDOM for SimKGC.[14]

We examine the structural characteristics on sets $S_m$ and $S_l$ of entities in 1, 000 most and least frequent triples, respectively, visited by BRWR. The entities in $S_m$ have an average degree of 11.1, compared to 297.3 for those in $S_l$. The betweenness centrality[15] averages around $5.2 \times 10^{-5}$ for $S_m$ and $8.2 \times 10^{-4}$ for $S_l$. These observations implies that SaaM prioritizes visiting unpopular and peripheral triples in KG over focusing on information-rich triples.

## 6 Analysis

### 6.1 Analysis on Negative Triples

Figure 7 shows how the cosine similarity distribution of in-batch negative triples varies depending on the epoch of SimKGC and SATKGC for FB15k-237. SATKGC encounters consistently more

hard negatives with scores from 0.2 to 1.0 than SimKGC, though the majority of the scores range from -0.2 to 0.2 by the end of training for both methods.[16] We speculate that SATKGC ends up with distinguishing positives from the hard negatives sampled from the subgraphs of the KG, as opposed to SimKGC which employs randomly-sampled easy negatives.

Based on our analysis, only 2.83% and 4.42% of the true triples ranked within the top 10 by SimKGC drop out of the top 10 for SATKGC on WN18RR and FB15k-237, respectively. In contrast, 34.87% and 13.03% of the true triples dropping out of the top 10 for SimKGC are ranked within the top 10 by SATKGC. This indicates that SATKGC effectively samples hard negatives while reducing false negatives.

An additional experiment in Appendix I shows how proximity-aware contrastive learning and frequency-aware mini-batch training selectively penalize hard negative triples. We also compare the effectiveness of using negatives fully sampled from SaaM and that partially sampled from SaaM in Appendix H.

### 6.2 Runtime Analysis

Running SATKGC incurs a marginal computational overhead, because (i) sampling subgraphs and (ii) computing distances and degrees are performed in advance before training. As shown in Table 7, the computational cost for (i) and (ii) is acceptable, and depends on the mini-batch size, which can be adjusted. Moreover, the time complexity for (ii) is acceptable. For each mini-batch $B$ of triples, we run Dijkstra's single source shortest path algorithm, and thus the runtime to compute the distance is O($|V|log|V| + |B|log|V|$), where $V$ is a set of entities in $B$.[17] Table 7 shows the training time per epoch for SATKGC, SimKGC [39], and StAR [38]. SATKGC remains competitive, though it takes slightly more time than SimKGC due to the computational overhead for (a) computing its loss using the shortest path length and the degree, (b) counting the occurrences of visited triples in SaaM, and (c) fetching subgraphs in SaaM.[18]

## 7 CONCLUSION

We propose a generic training scheme and new contrastive learning (CL) for KGC. Harmonizing SaaM using a subgraph of KG as a mini-batch, and both CL and mini-batch training incorporating the structural inductive bias of KG in fine-tuning PLMs helps learning the contextual text embeddings aware of the difficulty in the structural context of KG. Our findings imply that unequally feeding triples in training and leveraging the unique characteristics of KG lead to the effective text-based KGC method, achieving state-of-the-art performance on the three KG benchmarks.

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

**Figure 8: For each triple $(h, r, t)$, the bi-encoders take the name and description of $h$ and $t$ along with the text for $r$ as input.**

## A Input Format Details for Encoders

We adopt MPNet [31] as $Encoder_{hr}$ and $Encoder_t$, and set the maximum token length to 50. For each triple $h, r, t$, we concatenate the name of $h$, its textual description, and the name of $r$ with a special symbol [SEP] in between, and treat the concatenation as input for $Encoder_{hr}$, while the input for $Encoder_t$ consists the name and textual description of $t$. As illustrated in Figure 8, given a triple (Leonardo da Vinci, painted, Mona Lisa), the input to $Encoder_{hr}$ is the concatenation of [CLS], "Leonardo da Vinci", [SEP], the head's description, [SEP], and the relation's name "painted." The input to $Encoder_t$ is the concatenation of [CLS], "Mona Lisa", [SEP], and the tail description.

## B Inference

For inference, we calculate the cosine similarity between $\mathbf{x}_{hr}$ for a given $(h, r, ?)$ and $\mathbf{x}_t$ for all entities $t$. Then the tails with the top-k largest cosine similarities are answers. For a single pair, we need $|E|$ forward passes of $Encoder_t$ to obtain $\mathbf{x}_t$ for all entities $t$. Given a set $T$ of test triples, $2|T|$ forward passes of $Encoder_{hr}$ are required to get $\mathbf{x}_{hr}$ for every triple $(h, r, ?) \in T$ and its inverse triple $(t, r^{-1}, ?)$, thus resulting in $O(|E| + |T|)$ computation in total.

## C Markov Chain Monte Carlo Based Subgraph Sampling

Inspired by a negative sampling approach [46] based on Markov chain Monte Carlo (MCMC) in a general graph, we propose a new method to sample subgraphs from the KG. A negative sampling distribution should be positively but sublinearly correlated with the positive sampling distribution, which was validated by Yang et al. [46]. To include the entities close to a positive triple in the KG in negative triples, we define the sampling distribution $p_n$ of the negative tail $\hat{t}$ as : $p_n(\hat{t}|h, r) \propto p_d(\hat{t}|h, r)^{\alpha}, 0 < \alpha < 1, p_d(\hat{t}|h, r) = \frac{\cos(\mathbf{x}_{hr}, \mathbf{x}_{\hat{t}})}{\sum_{e \in E} \cos(\mathbf{x}_{hr}, \mathbf{x}_e)}$. where $\alpha$ is a parameter to stabilize the optimization process, $E$ is a set of entities in the KG, and $p_d$ is the sampling distribution of the positive tail. Calculating the normalization term in $p_d(\hat{t}|h, r)$ is time consuming and almost impossible. Therefore, we sample a negative tail $\hat{t}$ from $\tilde{p}_d = \cos(\mathbf{x}_{hr}, \mathbf{x}_{\hat{t}})$ by using the

---

**Algorithm 1:** MCMC-based Subgraph Sampling

**Input:** DFS path $D = \{t_1, t_2, ..., t_d\}$, proposal distribution $q$, number $k$ of negative samples, $burn\_in$ period

**Output:** negative tails

Initialize current negative node $x$ at random
$i \leftarrow 0, S \leftarrow \emptyset$
**for** *each triple $t$ in $D$* **do**
    Initialize $j$ as 0
    $h, r \leftarrow$ head, relation in triple
    **if** $i \leq burn\_in$ **then**
        $i \leftarrow i + 1$
        Sample an entity $y$ from $q(y|x)$
        Generate $r \in [0, 1]$ uniformly at random
        **if** $r \leq \min(1, \frac{\cos(\mathbf{x}_{hr}, \mathbf{x}_y)^{\alpha}}{\cos(\mathbf{x}_{hr}, \mathbf{x}_x)^{\alpha}} \frac{q(x|y)}{q(y|x)})$ **then**
            $x \leftarrow y$
    **else**
        **while** $j < k$ **do**
            Sample an entity $y$ from $q(y|x)$
            Add $y$ to $S$
            $x \leftarrow y$
            $j \leftarrow j + 1$

**return** $S$;

---

Metropolis-Hastings (M-H) algorithm, and randomly select a triple whose head is $\hat{t}$.

Algorithm 1 describes the process of sampling a subgraph by the M-H algorithm. To prepare a mini-batch of triples for SaaM, we traverse KG using depth-first search (DFS). From each entity $e \in E$, we perform DFS until we visit $d$ triples. For every visited triple along the DFS path, the triple inherits the probability distribution $\tilde{p}_d$ from the previous triple in the path, and $k$ negative tails $\hat{t}$ are sampled from this distribution. Then $\tilde{p}_d$ is updated. The proposal distribution $q$ is defined as a mixture of uniform sampling and sampling from the nearest $k$ nodes with the 0.5 probability each [46]. Both $d$ and $k$ above are hyperparameters. The $d$ triples in a DFS path, the sampled $d \times k$ triples, and their inverse triples compose a subgraph. We throw away the tails $\hat{t}$ extracted during the $burn\text{-}in$

| WN18RR | | | | |
|---|---|---|---|---|
| Size | MRR | Hits@1 | Hits@3 | Hits@10 |
| 512 | 0.689 | 0.610 | 0.737 | 0.828 |
| 1024 | **0.694** | **0.626** | **0.743** | **0.833** |
| 1536 | 0.690 | 0.622 | 0.741 | 0.828 |
| 2048 | 0.681 | 0.610 | 0.732 | 0.824 |

| FB15k-237 | | | | |
|---|---|---|---|---|
| Size | MRR | Hits@1 | Hits@3 | Hits@10 |
| 1024 | 0.337 | 0.250 | 0.366 | 0.519 |
| 2048 | 0.352 | 0.255 | 0.374 | 0.525 |
| 3072 | **0.370** | **0.279** | **0.403** | **0.550** |
| 4096 | 0.362 | 0.270 | 0.391 | 0.542 |

**Table 8: The results of investigating the model performance with respect to the batch size.**

period, and use the tails extracted after the period as the heads of the triples in the subgraph.

## D Datasets

In this paper, we use three KGC benchmarks. WN18RR is a sparse KG with a total of 11 relations and $\sim 41k$ entities. WN18RR is the dataset derived from WN18, consisting of relations and entities from WordNet [22]. WN18RR addresses the drawbacks of test set leakage by removing the inverse relation in WN18. FB15k-237 is a dense KG with 237 relations. Wikidata5M, a much larger KG than the others, provides transductive and inductive settings. Wikidata5M-Trans is for the transductive setting, where entities are shared and triples are disjoint across training, validation, and test. Wikidata5M-Ind is for the inductive setting, where the entities and triples are mutually disjoint across training, validation, and test [40].

## E Implementation Details

In our weighted InfoNCE loss, additive margin $\gamma$ is set to 0.02. We select the best performing batch sizes of 1024 from {512, 1024, 1536, 2048} for WN18RR, and Wikidata5M, and 3072 from {1024, 2048, 3072, 4096} for FB15k-237. We set the restart probability $p_r$ to 1/25 in BRWR. We used six A6000 GPUs and 256G RAM. Training on WN18RR took 50 epochs, for a total of 4 hours. FB15k-237 took 30 epochs and a total of 3 hours, while Wikidata5M took 2 epochs and 19 hours.

## F Hyperparameter Sensitivity

We investigate how restart probability $p_r$ in BRWR affects model performance. The hyperparameter $p_r$ is associated with the length of the random walk path from the start entity, which in turn influences the occurrence of duplicate entities in a mini-batch. A longer path leads to fewer duplicate entities in the mini-batch. Figure 9(a) illustrates that a lower $p_r$ value, encouraging a longer random walk path, leads to higher Hits@1 for WN18RR and FB15k-237. We analyze the impact of duplicate entities in a mini-batch on the model performance. In Figure 9(b), more duplicate entities resulting from higher $p_r$ negatively impact on the performance, which highlights

| WN18RR | | | | |
|---|---|---|---|---|
| Method | MRR | Hits@1 | Hits@3 | Hits@10 |
| RotatE[32] | 0.471 | 0.421 | 0.490 | 0.568 |
| RotatE+SaaM | **0.479** | **0.433** | **0.505** | **0.580** |
| StAR[38] | 0.398 | 0.238 | 0.487 | 0.698 |
| StAR+SaaM | **0.411** | **0.261** | **0.511** | **0.729** |

| FB15k-237 | | | | |
|---|---|---|---|---|
| Method | MRR | Hits@1 | Hits@3 | Hits@10 |
| RotatE[32] | 0.335 | 0.243 | 0.374 | 0.529 |
| RotatE+SaaM | **0.343** | **0.255** | **0.389** | **0.534** |
| StAR[38] | 0.288 | 0.195 | 0.313 | 0.480 |
| StAR+SaaM | **0.319** | **0.220** | **0.334** | **0.490** |

**Table 9: Performance comparison between original StAR and StAR+SaaM where StAR+SaaM stands for the StAR model architecture trained by our training framework SaaM.**

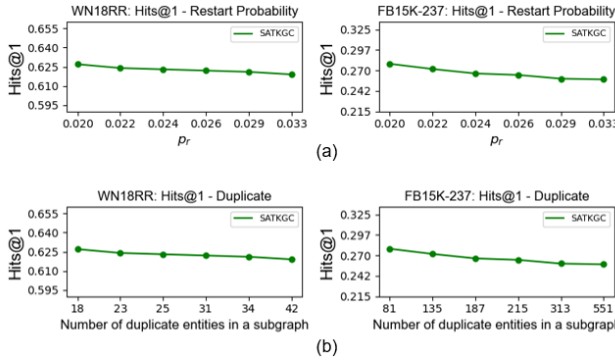

**Figure 9: (a) Impact of varying restart probabilities on the model performance. (b) Impact of varying the number of duplicate entities in a mini-batch on the model performance.**

the importance of reducing the duplicates in a mini-batch to avoid the performance degradation. Additionally, we compare the performance of SATKGC on WN18RR and FB15k-237 with varying batch sizes in Table 8. Based on the results, we select the batch size that produce the best performance as the default setting for SATKGC.

## G Performance of the SaaM Scheme Across Models

To demonstrate the generality of the SaaM scheme, we conduct additional experiments by applying the SaaM approach to RotatE, an embedding-based method, and StAR, a text-based method, to evaluate their performance on the WN18RR and FB15k-237 datasets. As shown in Table 9, incorporating the SaaM approach into both RotatE and StAR results in significant improvements across all metrics. For instance, on the FB15k-237, StAR + SaaM exhibits a significant improvement in Hits@1, increasing by 12.8%, from 0.195 to 0.220. These findings illustrate that SaaM is model-agnostic and highly generalizable, effectively enhancing performance regardless of the underlying model.

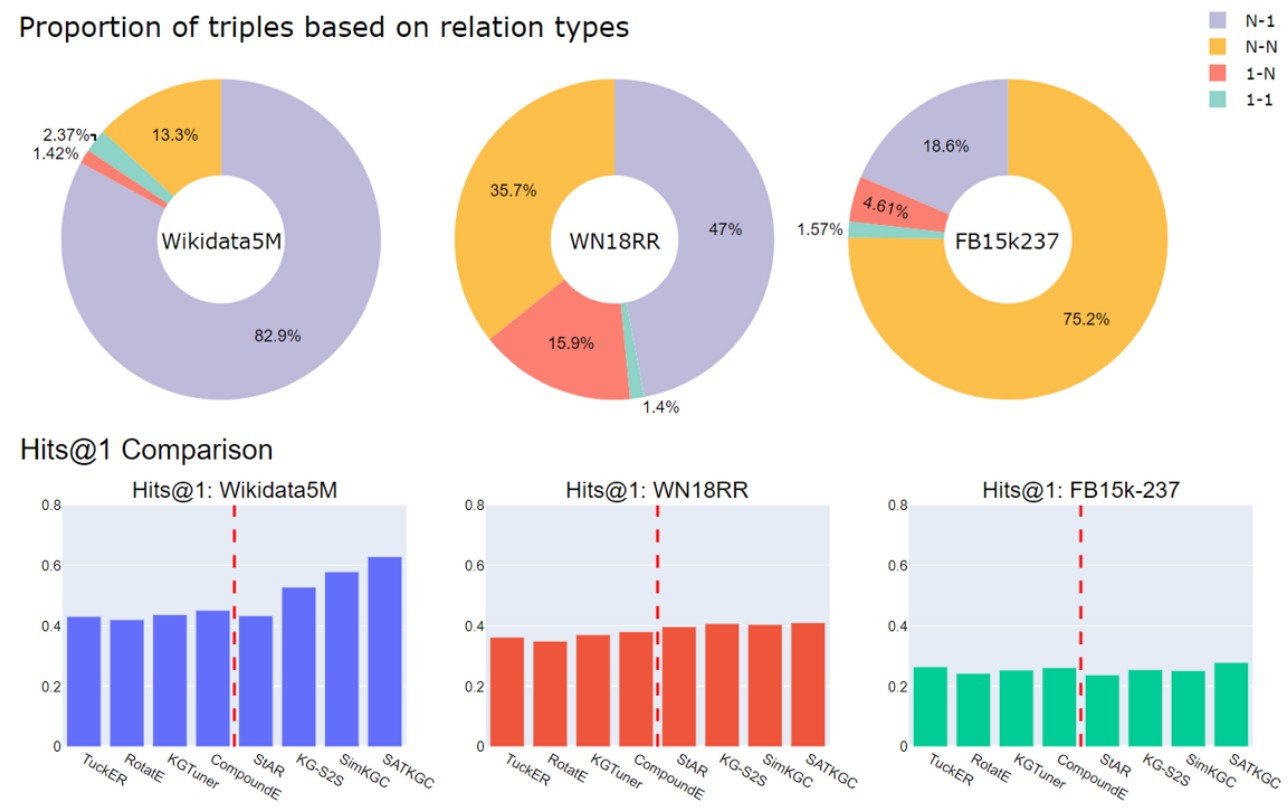

**Figure 10: The proportion of triples categorized by relation types for the Wikidata5M-Trans, WN18RR, and FB15k-237 datasets, along with their corresponding Hits@1 performance results.**

| WN18RR | | | | |
|---|---|---|---|---|
| Method | MRR | Hits@1 | Hits@3 | Hits@10 |
| Mixed | 0.676 | 0.610 | 0.729 | 0.816 |
| SaaM | **0.694** | **0.626** | **0.743** | **0.833** |

| FB15k-237 | | | | |
|---|---|---|---|---|
| Method | MRR | Hits@1 | Hits@3 | Hits@10 |
| Mixed | 0.349 | 0.253 | 0.393 | 0.538 |
| SaaM | **0.370** | **0.279** | **0.403** | **0.550** |

**Table 10: Performance comparison between Mixed and SaaM, where Mixed replaces half of in-batch triples generated by SaaM with randomly selected triples.**

| Triples | $\psi_{\mathcal{G}}(t)\mathcal{L}_{(h,r,t)}$ | $\mathcal{L}_{(h,r,t)}$ |
|---|---|---|
| total triples | 0.2268 | 0.1571 |
| false positives | **0.4987** | **0.2834** |

**Table 11: Comparison of average loss values on WN18RR for total triples and false positives in the batch.**

## H  Performance of Hybrid In-Batch Negatives

To evaluate the efficacy of constructing a batch exclusively from triples in a subgraph (i.e., SaaM), we compare two sampling methods: SaaM and Mixed. Mixed is a variant that replaces half of the triples in each mini-batch generated by SaaM with randomly selected triples. Table 10 illustrates the performance of SaaM and Mixed. Including randomly selected triples degrades performance, as evidenced by a significant drop in Hits@1, indicating that a mini-batch composed solely of triples within the subgraph is more beneficial.

## I  False Positive Analysis

We aim to demonstrate that our contrastive learning with two structure-aware factors selectively penalizes hard negative triples. Table 11 compares $\psi_{\mathcal{G}}(t)\mathcal{L}_{(h,r,t)}$ and $\mathcal{L}_{(h,r,t)}$ on average, specifically for false positives of the existing state-of-the-art model [39] and for all training triples. In this comparison, $\mathcal{L}_{(h,r,t)}$ in Equation (3) represents the loss with only a structural hardness factor $\omega_{ht_i}$ applied, while $\psi_{\mathcal{G}}(t)\mathcal{L}_{(h,r,t)}$ in Equation (5) additionally incorporates a reweighting factor $\psi_{\mathcal{G}}(t)$. The average loss values for the false positives are higher than those for all the triples, which indicates that the structure-aware contrastive learning method severely punishes incorrect predictions for hard negative triples.

| Approach | WN18RR | | | | FB15k-237 | | | |
|---|---|---|---|---|---|---|---|---|
| | MRR | Hits@1 | Hits@3 | Hits@10 | MRR | Hits@1 | Hits@3 | Hits@10 |
| *Embedding-based approach* | | | | | | | | |
| SANS [1] | 0.216 | 0.027 | 0.322 | 0.509 | 0.298 | 0.203 | 0.331 | 0.486 |
| TransE [3] | 0.239 | 0.421 | 0.450 | 0.510 | 0.280 | 0.193 | 0.372 | 0.439 |
| DistMult [44] | 0.435 | 0.410 | 0.450 | 0.510 | 0.280 | 0.195 | 0.297 | 0.441 |
| TuckER [2] | 0.466 | 0.432 | 0.478 | 0.518 | 0.361 | 0.265 | 0.391 | 0.538 |
| RotatE [32] | 0.471 | 0.421 | 0.490 | 0.568 | 0.335 | 0.243 | 0.374 | 0.529 |
| KGTuner [53] | 0.481 | 0.438 | 0.499 | 0.556 | 0.345 | 0.252 | 0.381 | 0.534 |
| KG-Mixup [30] | 0.488 | 0.443 | 0.505 | 0.541 | 0.359 | 0.265 | 0.395 | 0.547 |
| UniGE[21]† | 0.491 | 0.447 | 0.512 | 0.563 | 0.343 | 0.257 | 0.375 | 0.523 |
| CompoundE [10] | 0.492 | 0.452 | 0.510 | 0.570 | 0.350 | 0.262 | 0.390 | 0.547 |
| KRACL [34] | 0.529 | 0.480 | 0.539 | 0.621 | 0.360 | 0.261 | 0.393 | 0.548 |
| CSProm-KG [5] | 0.569 | 0.520 | 0.590 | 0.675 | 0.355 | 0.261 | 0.389 | 0.531 |
| *Text-based approach* | | | | | | | | |
| KG-BERT [47] | 0.216 | 0.040 | 0.298 | 0.516 | 0.158 | 0.019 | 0.232 | 0.420 |
| MTL-KGC [18] | 0.331 | 0.203 | 0.383 | 0.597 | 0.267 | 0.172 | 0.298 | 0.458 |
| StAR [38] | 0.398 | 0.238 | 0.487 | 0.698 | 0.288 | 0.195 | 0.313 | 0.480 |
| HaSa [50] | 0.535 | 0.446 | 0.587 | 0.711 | 0.301 | 0.218 | 0.325 | 0.482 |
| KG-S2S [4] | 0.572 | 0.529 | 0.595 | 0.663 | 0.337 | 0.255 | 0.374 | 0.496 |
| SimKGC [39] | 0.671 | 0.580 | 0.729 | 0.811 | 0.340 | 0.252 | 0.365 | 0.515 |
| GHN [27]‡ | 0.678 | 0.596 | 0.719 | 0.821 | 0.339 | 0.251 | 0.364 | 0.518 |
| *Ensemble approach* | | | | | | | | |
| StAR(Self-Adp) [38] | 0.520 | 0.456 | 0.509 | 0.707 | 0.332 | 0.229 | 0.387 | 0.526 |
| SATKGC (*w/o SaaM*) | 0.673 | 0.595 | 0.728 | 0.813 | 0.349 | 0.256 | 0.367 | 0.520 |
| SATKGC (*w/o PCL, FMT*) | 0.676 | 0.608 | 0.722 | 0.820 | 0.355 | 0.261 | 0.386 | 0.537 |
| SATKGC (*w/o FMT*) | 0.680 | 0.611 | 0.729 | 0.823 | 0.351 | 0.265 | 0.389 | 0.541 |
| SATKGC (*w/o PCL*) | 0.686 | 0.623 | 0.740 | 0.827 | 0.366 | 0.272 | 0.396 | 0.545 |
| SATKGC | **0.694** | **0.626** | **0.743** | **0.833** | **0.370** | **0.279** | **0.403** | **0.550** |

**Table 12: KGC results for the WN18RR, and FB15k-237 datasets. "PCL" and "FMT" refer to Proximity-aware Contrastive Learning and Frequency-aware Mini-batch Training respectively. The best and second-best performances are denoted in bold and underlined respectively. †: numbers are from Liu et al. [21] ‡: numbers are from Qiao et al. [27].**

## J  Correlation of Relation Types and Performances

We investigate the distribution of triples based on their relation types on Wikidata5M-Trans, WN18RR, and FB15k-237. Figure 10 shows that the proportion of triples with the N-N relation type increases in the order of Wikidata5M-Trans, WN18RR, and FB15k-237, while the proportion of triples with the N-1 relation type decreases in the same order. In Table 2 and Table 3, We observe that text-based methods outperform embedding-based methods in Wikidata and WN18RR, which have a higher proportion of N-1 relation type and lower proportion of N-N relation type. In contrast, in FB15k-237, which has a higher proportion of N-N relation type and a lower proportion of N-1 relation type, embedding-based methods generally achieve better performance than text-based methods. This performance difference between the datasets is larger for a text-based approach than for an embedding-based approach. We speculate that this is because the embedding-based approach randomly initializes the entity and relation embeddings, while the text-based approach uses contextualized text embeddings obtained from PLMs. For the N-N relations where multiple tails can be the correct answer for the same $(h, r)$ pair, the embeddings of these correct tails should be similar. However, PLMs take only text as input, being oblivious of their high similarity. Therefore, these tail embeddings generated

| Wikidata5M-Trans | | | | |
|---|---|---|---|---|
| Model | MRR | Hits@1 | Hits@3 | Hits@10 |
| ReSKGC | 0.396 | **0.373** | 0.413 | 0.437 |
| SATKGC | **0.408** | 0.366 | **0.425** | **0.479** |

**Table 13: Performance comparison between SATKGC and ReSKGC. ReSKGC is a retrieval-augmented generation(RAG)-based model that performs search on the knowledge graph rather than learning the knowledge graph itself.**

by the PLMs might be far apart from each other, so the $(h, r)$ embedding is likely to remain in the middle of these tail embeddings during fine-tuning.

## K  Entire Main Results

Table 12 presents the results of all baselines compared with ours.

## L  Peformance Comparison with RAG-based Model

We additionally conduct performance comparison experiments with the RAG-based model ReSKGC [48]. Since ReSKGC does not have publicly available source code, we use the performance results reported directly in the paper. Furthermore, as there are no reported results for FB15k-237 and WN18RR, we only present the results on

Wikidata5M-Trans. As shown in Table 13, SATKGC demonstrates superior performance across all metrics except for Hits@1. Unlike SATKGC, ReSKGC is a decoder-based generation model. When given the head and relation IDs, it converts them into text, retrieves the most semantically relevant triples from the KG using a retrieval component such as BM25, and generates the correct tail based on that information. Therefore, unlike SATKGC, ReSKGC does not learn the KG directly and only uses it for retrieval, without utilizing any structural information of the KG. This explains why SATKGC achieves overwhelmingly better performance, despite both models having the same number of parameters used during training.

Received 14 October 2024
