# OpenReview forum: "Subgraph-Aware Training of Language Models for Knowledge Graph Completion Using Structure-Aware Contrastive Learning"
_ACM.org/TheWebConf/2025/Conference — WWW 2025 Oral_

### Official Review · Reviewer_tXuk · 2024-11-26

**Novelty:** 3
**Technical Quality:** 5

**Review:**

This paper proposed a Subgraph-Aware Training framework. The core of the article is the method of sub batch sampling and the construction method of subgraphs. The baseline model used is similar to the previously published comparative learning article 'Language Models as Knowledge Embeddings‘.
### pros:
1. The paper is written very clearly
2. The theoretical part is relatively sufficient
### cons:
1. The score of the model is not high enough and lacks comparison with recent papers
2. Some formulas in the paper are not labeled

**Questions:**

1. For the FB15K-237 dataset, there are currently many methods with higher scores, such as [1] Language Models as Knowledge Embeddings. [2] Relation Prediction as an Auxiliary Training Objective for Improving Multi-Relational Graph Representations
2. The overall framework of the model is relatively simple, and the framework of the comparative model is still modeled after [1], which lacks innovation.

**Reviewer Confidence:**

4: The reviewer is certain that the evaluation is correct and very familiar with the relevant literature

**Scope:**

3: The work is somewhat relevant to the Web and to the track, and is of narrow interest to a sub-community

---

### Official Review · Reviewer_4tPq · 2024-12-01

**Novelty:** 5
**Technical Quality:** 6

**Review:**

This paper proposes the **Subgraph-Aware Training for Knowledge Graph Completion**,  fine-tuning PLMs by incorporating the structural properties of knowledge graphs (KGs) to address challenges like long-tailed entity distributions and hard-to-distinguish negative samples. SATKGC uses biased random walks to sample subgraphs as training batches, ensuring equal representation of entities, and employs proximity-aware contrastive learning to weight negative samples based on their graph proximity, emphasizing harder negatives.  Besides, it reweights the training loss to prioritize low-frequency entities, aligning mini-batch distributions with the original KG.

This paper is well-written, and its motivation is strong. The proposed method is simple yet effective, and the experiments are comprehensive, demonstrating the method's effectiveness across multiple benchmarks.

**Questions:**

- How does the subgraph sampling strategy scale with extremely large knowledge graphs, and what are the trade-offs in terms of time and memory usage?

- Why is proximity (shortest path) chosen as the primary metric for identifying hard negatives? Are there other structural metrics, such as betweenness centrality or clustering coefficient, offering additional insights? If the graph structure is highly connected or extremely sparse, would this method handle such scenarios?

- How can we know the model captures both the structural and textual properties of the graph?

**Reviewer Confidence:**

3: The reviewer is confident but not certain that the evaluation is correct

**Scope:**

4: The work is relevant to the Web and to the track, and is of broad interest to the community

---

### Official Review · Reviewer_8q3y · 2024-12-03

**Novelty:** 5
**Technical Quality:** 6

**Review:**

In this paper, the authors propose a model to predict link in knowledge graphs based on text encoders, contrastive learning and subgraph-aware mini-batching and negative sampling.

Overall, the paper is well-written, easy to follow. The issues (skewed long tail distribution of KG) is well-known and of importance. The different modules are well motivate and grounded in different lines of research from the literature. Experiments appear sound and extensive, showcasing the improvement brought by each proposed module, comparing with other models on different settings (different encoders, transductive and inductive). I also appreciate the effort put by the authors to illustrate the skewed distributions of degrees, and distance between entities, and their impact on false positive. It is also noteworthy that the code is publicly available, ensuring reproducibility.

I only have a few remarks:
- It is not clear in the introduction that when mentioning  "closeness between tail and head of a false triple" you actually refer to the number of hops in the graph. I was initially thinking of distance in the embedding space. This should be clarified.
- You mention targeting Knowledge Graph Completion, but this appears more of Link Prediction to me, as KGC also includes other tasks (e.g., type prediction)
- The state of the art is not mentioning all the approaches for link prediction based on rules
- Section 3 "x_hr for h and t" should be "x_hr for h and r"
- In Section 4.2, "we randomly select |B|/2 triples from the subgraph", why does this selection occur?

**Questions:**

I only have one question:
- In Section 4.2, "we randomly select |B|/2 triples from the subgraph", why does this selection occur?

**Reviewer Confidence:**

3: The reviewer is confident but not certain that the evaluation is correct

**Scope:**

3: The work is somewhat relevant to the Web and to the track, and is of narrow interest to a sub-community

---

### Official Review · Reviewer_oWwK · 2024-12-03

**Novelty:** 5
**Technical Quality:** 6

**Review:**

This article presents a novel approach for improving knowledge graph completion based on three main techniques: (i) to sample subgraphs of the KG and treating triples from each subgraph as mini-batches; (ii) fine-tuning of pre-trained language models by focusing on negative triples; and (iii) use of a loss function to mitigate the imbalanced nature of knowledge graphs. Those techniques have shown to be promising when it comes to performing knowledge graph completion. To further prove this, the authors have shown that the topological structure of knowledge graphs, e.g. shortest, subgraphs, and degree, is strongly related to the performance of PLM-based KGC methods. This leads to a reduction of the false positives when compared to existing methods.

In general, this article covers a wide range of contributions, including literature review, a new approach for knowledge graph completion using topological structures of the graph and extensive experiments on three KGC benchmarks to compare the proposed algorithm against existing methods. The ablation studies presented confirm the contributions of each individual component. While the description of the proposed methods is sufficient, it lacks depth in certain areas, particularly in formal definitions (preliminaries) and the bridge between methodological novelty and practical implementation could be improved; e.g. it does not provide enough explanations about how these methods are implemented and how they affect performance or efficiency of the proposed solution. As an example, this paper mentions scalability but does not analyze computational trade-offs in depth: how does subgraph sampling (to say one proposed method) impact training times when compared to simpler baselines? Are there any possible optimizations that could improve the training times? In relation to the implementation details, there is limited detail on how these methods are integrated into a training pipeline. As such, it is hard to reproduce the experiments without any additional detail.

## Detailed comments

- Page 1: “A negative triple difficult for a KGC method to distinguish from its corresponding positive triple is regarded as a hard negative triple.” I find this sentence difficult to follow, I think it should be reviewed for improving the clarity.

- Page 8, Section 6.2: I find that this section should be expanded, including some descriptions of the trade-offs between efficiency and accuracy. I would also like to know how the proposed solution compares to the baselines in terms of memory efficiency.

- Page 8. Section 6.2: is it possible to create a lighter version of SATKGC for resource-constrained environments? I would like to see a discussion of potential trade-offs between model performance and computational complexity.

- Page 9, Section 7: I would like to see some of the limitations that the authors have encountered, as well as the future work, e.g. how the proposed method will continue to evolve: is everything that could be done already done? One could claim that SATKGC could be applied to other domains, including temporal KGs…

- Appendix F: I would like to see a further discussion of hyperparameter tuning based on both: more datasets, and also exploring other of the hyperparameters, e.g. temperature parameter in contrastive loss.

- I would like to see an example of training a KGC model with SATKGC so that readers could use the tool. Even if it is true that one can follow the instructions in the repository, I believe that a short description of the implementation details could be beneficial.

- This paper is a bit more general than the scope of the conference, as it does not mention how it is beneficial for the web. It mentions graphs, describing how it can be used for performing knowledge graph completion, but it does not mention how it can be beneficial for the web. It does not mention any web-related technology either, e.g. RDF: the formal definitions of graphs are not RDF-based graphs. However, I see that this paper is an application of learning in graphs with missing information, and that is - indeed - helpful for the web. Thus, the issue is just that it is not clarified by the authors how this could be useful or related to the web.

**Questions:**

- Is it possible to create a lighter version of SATKGC for resource-constrained environments?
- Could you elaborate about the tradeoffs and assess the pros and cons of this approach?

**Reviewer Confidence:**

1: The reviewer's evaluation is an educated guess

**Scope:**

2: The connection to the Web is incidental, e.g., use of Web data or API